# Novel Organization of the Staphylococcal Cassette Chromosome *mec* Composite Island in Clinical *Staphylococcus haemolyticus* and *Staphylococcus hominis* Subspecies *hominis* Isolates from Dogs

Nathita Phumthanakorn,[a,e] Thidathip Wongsurawat,[b] Piroon Jenjaroenpun,[b] Alongkorn Kurilung,[c] Nuvee Prapasarakul[d,e]

[a]Department of Pre-Clinic and Applied Animal Science, Faculty of Veterinary Science, Mahidol University, Nakhon Pathom, Thailand

[b]Division of Bioinformatics and Data Management for Research, Research Group and Research Network Division, Faculty of Medicine Siriraj Hospital, Mahidol University, Bangkok, Thailand

[c]Siriraj Metabolomics and Phenomics Center, Faculty of Medicine Siriraj Hospital, Mahidol University, Bangkok, Thailand

[d]Department of Veterinary Microbiology, Faculty of Veterinary Science, Chulalongkorn University, Bangkok, Thailand

[e]Center of Excellence in Diagnostic and Monitoring of Animal Pathogens, Chulalongkorn University, Bangkok, Thailand

**ABSTRACT** *Staphylococcus haemolyticus* and *Staphylococcus hominis* subsp. *hominis* are common coagulase-negative *staphylococcus* opportunistic pathogens. In Thailand, the clinical strains *S. haemolyticus* 1864 and 48 and *S. hominis* subsp. *hominis* 384 and 371 have been recovered from sick dogs. These strains were methicillin resistant with the nontypeable staphylococcal cassette chromosome *mec* (NT-SCC*mec*). The SCC*mec* element distribution in the clinical isolates from dogs was analyzed using whole-genome sequencing, which revealed the presence of different SCC*mec* composite islands (CIs) and gene structure. The SCC*mec*-CIs of $\psi$SCC*mec*$_{1864}$ (13 kb) and $\psi$SCC$_{1864}$ (11 kb) with a class C1 *mec* complex but no *ccr* gene were discovered in *S. haemolyticus* 1864. The CIs of $\psi$SCC*mec*$_{48}$ with a C1 *mec* complex (28 kb), SCC$_{48}$ with *ccrA4B4* (23 kb), and $\psi$SCC$_{48}$ (2.6 kb) were discovered in *S. haemolyticus* 48. In SCC$_{48}$, insertion sequence IS*256* contained an aminoglycoside-resistant gene [*aph(2")-Ia*]. Two copies of IS*431* containing the tetracycline-resistant gene *tet*(K) were found downstream of $\psi$SCC$_{48}$. In *S. hominis* subsp. *hominis*, the SCC*mec*-CI in strain 384 had two separate sections: $\psi$SCC*mec*$_{384}$ (20 kb) and SCC$_{ars}$ (23 kb). $\psi$SCC*mec*$_{384}$ lacked the *ccr* gene complex but carried the class A *mec* complex. Trimethoprim-resistant dihydrofolate reductase (*dfrC*) was discovered on $\psi$SCC*mec*$_{384}$ between two copies of IS*257*. In strain 371, SCC*mec* VIII (4A) (37 kb) lacking a direct repeat at the chromosomal end was identified. This study found SCC*mec* elements in clinical isolates from dogs that were structurally complex and varied in their genetic content, with novel organization.

**IMPORTANCE** In Thailand, the staphylococcal cassette chromosome *mec* (SCC*mec*) element, which causes methicillin resistance through acquisition of the *mec* gene, has been studied in clinical coagulase-negative *Staphylococcus* isolates from various companion animals, and *Staphylococcus haemolyticus* and *Staphylococcus hominis* subsp. *hominis* were found to have the most nontypeable (NT)-SCC*mec* elements. These species are more prone to causing illness and more resistant to a variety of antimicrobials than other coagulase-negative staphylococci. However, full characterization of NT-SCC*mec* in clinical *S. haemolyticus* and *S. hominis* subsp. *hominis* isolates from such animals has been limited. Our findings support the use of full nucleotide sequencing rather than PCR designed for *Staphylococcus aureus* in further research of novel SCC*mec* elements. Moreover, several antimicrobial resistance and heavy metal resistance genes were identified on the SCC*mec* elements; these are important as they could limit the therapeutic options available in veterinary medicine.

Address correspondence to Nuvee Prapasarakul, nuvee.p@chula.ac.th.

The authors declare no conflict of interest.

**KEYWORDS** *Staphylococcus haemolyticus*, *Staphylococcus hominis* subsp. *hominis*, SCC*mec*, composite island, dog

Antimicrobial resistance—specifically, methicillin-resistant staphylococci (MRS)—is a major problem in humans and animals. The acquisition of the *mec* gene on the staphylococcal cassette chromosome *mec* (SCC*mec*) element confers methicillin resistance, which has been linked to multiple antimicrobial resistance events (1). SCC*mec* elements are unique mobile genetic elements that encode resistance to methicillin and almost all $\beta$-lactam antibiotics; they integrate into the staphylococcal chromosomal attachment site (*attB*) within the *orfX* gene (2, 3). The *mec* gene complex, cassette chromosome recombinase (*ccr*) gene complex, and adjoining region are the core structural and critical genetic components shared by SCC*mec* elements (3). SCC*mec* structures are highly complex, and a wide range of sizes are found in different *Staphylococcus* species (1). Horizontal *mec* and SCC gene transfer is known to be possible between coagulase-negative staphylococci (CoNS) and *Staphylococcus aureus*, implying that CoNS could be a source and reservoir for SCC*mec* (4–6). The transfer of genetic materials within SCC*mec* between human and animal strains, including as antimicrobial resistance genes and CRISPR/Cas system genes, was also detected (7, 8). Furthermore, SCC*mec* components are more common and diverse in CoNS from animals than those in *S. aureus*, although the overall structure has yet to be described (3, 9, 10). To increase our understanding of SCC*mec* evolution and expand the nomenclature of SCC*mec* types in CoNS, it will be necessary to fully characterize a unique and/or subtype of SCC*mec* structure in CoNS isolates from animals (3).

In Thailand, the SCC*mec* element was previously studied in clinical CoNS isolates from dogs and cats (11). The most nontypeable (NT) SCC*mec* elements were found in *Staphylococcus haemolyticus* and *S. hominis* subsp. *hominis*, which were negative detection of the *mec* and/or *ccr* gene complex using multiplex PCR (11). Furthermore, they are more prone to causing illness and more resistant to various antimicrobials than are other CoNS (12). To date, full characterization of NT-SCC*mec* in clinical *S. haemolyticus* and *S. hominis* subsp. *hominis* isolates from animals is limited. Therefore, in this study, a combination of short- and long-read sequencing was used to examine the entire structure of SCC*mec* elements in clinical *S. haemolyticus* and *S. hominis* subsp. *hominis*, which could not be identified using multiplex PCR. Specifically, the structural and genetic content of novel SCC*mec* composite islands (CIs) was analyzed with the aim of elucidating the mechanisms by which SCC*mec* is acquired in the genome.

## RESULTS

**SCC*mec* containing the *mec* complex but lacking *ccr* in *S. haemolyticus* 1864.** In *S. haemolyticus* 1864, the CIs of $\psi$SCC*mec*$_{1864}$ (13 kb) and $\psi$SCC$_{1864}$ (11 kb) were discovered. The $\psi$SCC*mec*$_{1864}$ element was located immediately downstream of *orfX* and was flanked by direct repeats (DRs) and their inverted repeat (IR) sequences (DR1-IR1 and DR2-IR2) (Fig. 1A). At this position, the class C1 *mec* complex of insertion sequence IS*431*-*mecA*-$\Delta$*mecR1*-IS*431*, with two copies of IS*431* orientated in the same direction, was discovered. The *ccr* gene was not discovered. The genes *cadC*, *arsC*, *arsB*, and *copA*, which encode cadmium, arsenic, and copper resistance, respectively, were found in $\psi$SCC*mec*$_{1864}$ (Fig. 1A; see also Table S1 in the supplemental material). The second region of $\psi$SCC$_{1864}$ was found immediately downstream from $\psi$SCC*mec*$_{1864}$ and flanked by DR2-IR2 and IR3. DR3 was not found at the chromosomal end of this element. The IS*Sep1*, *cadC*, *arsC*, and *arsB* genes were found on $\psi$SCC$_{1864}$.

The SCC*mec*-CIs in *S. haemolyticus* 1864 were nearly identical (99.7% nucleotide similarity) to those of the SCC*mec* in *S. haemolyticus* Sh29/312/L2 (GenBank accession number CP011116), isolated from a human patient (13) (Fig. 1A). Detailed information on SCC*mec* in *S. haemolyticus* Sh29/312/L2 has not yet been determined. In contrast, the additional IS*256* and IS*Sha1* insertion sequences were located at the chromosomal region in *S. haemolyticus* 1864 but not in *S. haemolyticus* Sh29/312/L2. $\psi$SCC*mec*$_{1864}$ showed high sequence similarity (>96%) to the previously described $\psi$SCC*mec*

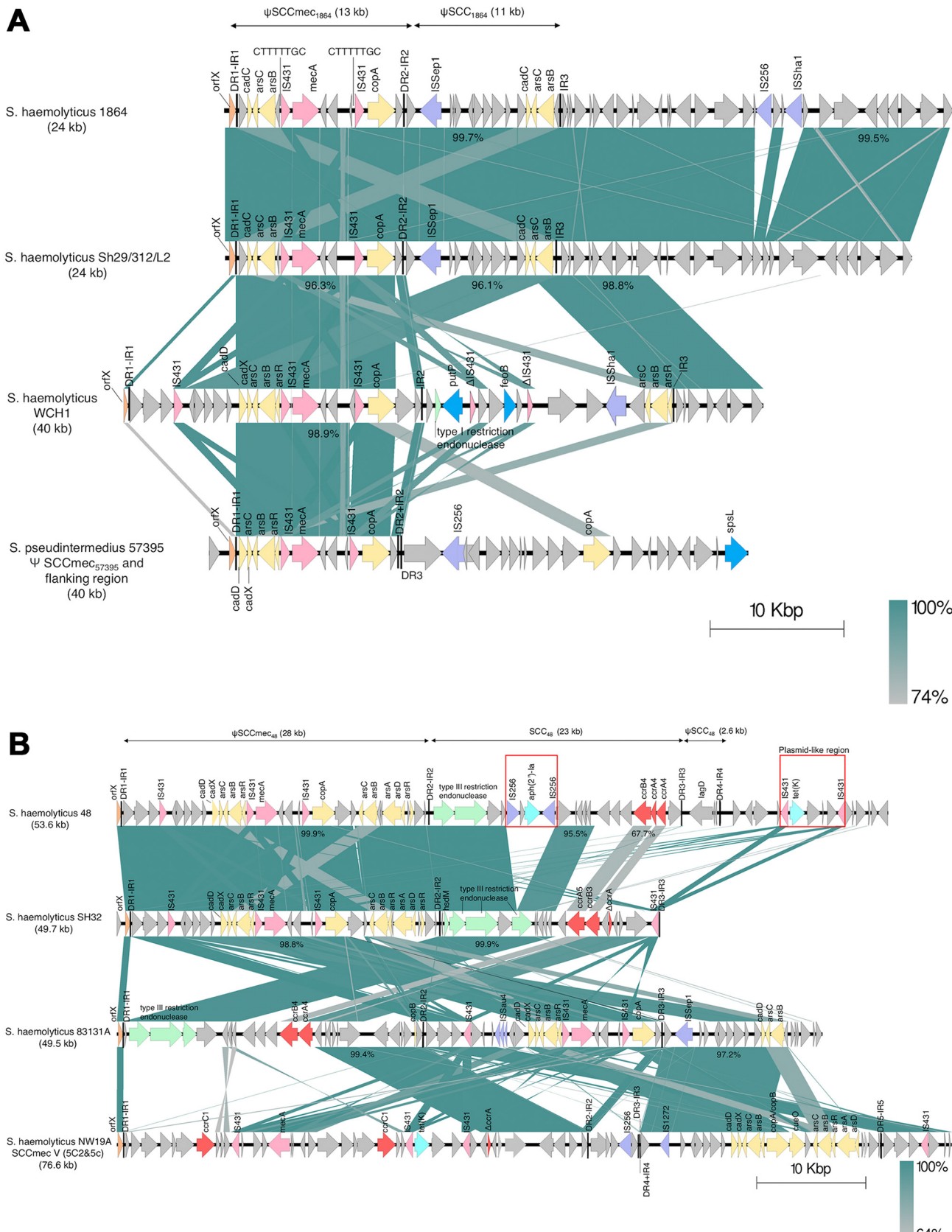

**FIG 1** Comparative analysis of the structural elements in the staphylococcal cassette chromosome *mec* composite islands (SCC*mec*-CI) in (A) *S. haemolyticus* 1864 (GenBank accession number JAKUUX000000000), *S. haemolyticus* Sh29/312/L2 (CP011116), *S. haemolyticus* WCH1 (JQ764731), and

element in *S. haemolyticus* WCH1 (JQ764731.1), obtained from the blood of a patient in China (14), and to $\psi$SCC*mec*$_{57395}$ (HE984157.2) in an *S. pseudintermedius* clinical isolate from a dog (15) (Fig. 1A). Similar to WCH1, identical 8-bp sequences (-CTTTTTGC-) were discovered adjacent to two copies of IS*431* in $\psi$SCC*mec*$_{1864}$. The genetic characteristics and highest gene homology of SCC*mec* in 1864 are listed in Table S1.

**SCC*mec*-CI of $\psi$SCC*mec*$_{48}$, SCC$_{48}$, and $\psi$SCC$_{48}$ in *S. haemolyticus* 48.** The SCC*mec*-CI element in *S. haemolyticus* 48 comprised 53.6 kp and was flanked by DR1-IR1 and DR4-IR4. Two additional DRs and IRs (DR2-IR2 and DR3-IR3) were found within the element, which was divided into three parts: $\psi$SCC*mec*$_{48}$ (28 kb), SCC$_{48}$ (23 kb), and $\psi$SCC$_{48}$ (2.6 kb) (Fig. 1B). The first $\psi$SCC*mec*$_{48}$ region contained a class C1 *mec* complex of IS*431*-*mecA*-$\Delta$*mecR1*-IS*431* and two copies of IS*431* organized in the same orientation. The genes *cad*, *ars*, and *cop* were discovered upstream and downstream of the *mec* gene complex. The second SCC$_{48}$ region contained two copies of *ccrA4* (477 and 882 bp) and one of *ccrB4* (1,626 bp). The two copies of *ccrA* exhibited 99.79% (477 bp) and 99.32% (882 bp) nucleotide similarity, respectively, to the *ccrA4* gene of *S. aureus* M06/0171 (GenBank accession number HE980450.1), whereas *ccrB* had 100% similarity to *ccrB4* of *S. haemolyticus* 06 (KX524951.1) (Fig. 1B and Table S2). In addition, *aph(2")-Ia*, an aminoglycoside resistance gene, was discovered on the plasmid-like region of IS*256*-*aph(2")-Ia*-IS*256*. The *aph(2")-Ia* sequence was 100% identical to that in *S. aureus* UMCG578 (CP077738.1), *Enterococcus faecalis* E533 (CP086726.1), and *Staphylococcus lugdunensis* CGMH-SL118 (CP048008.1). The third chromosomal end of the $\psi$SCC$_{48}$ region harbored lactococcin G-processing and transport ATP-binding protein (*lagD*) and 2 hypothetical protein genes. A plasmid-like region of IS*431* harboring the tetracycline-resistant gene *tet*(K) was discovered adjacent to $\psi$SCC$_{48}$ (Fig. 1B). The *tet*(K) gene had the highest nucleotide similarity to the gene sequences of *S. saprophyticus* UTI-056 plasmid pUTI-056-3 (100% similarity) and *S. haemolyticus* NW19A (99.85% similarity).

The gene homology and organization of $\psi$SCC*mec*$_{48}$ were highly identical (99.9%) to those of the SCC*mec* element in *S. haemolyticus* SH32 (GenBank accession number KF006347.1), a clinical isolate from a human (5) (Fig. 1B). There were differences in the types of *ccr* gene complexes, with *S. haemolyticus* 48 having *ccrA4B4* and *S. haemolyticus* SH32 having *ccrA5B3*. Furthermore, regions containing *aph(2")-Ia* and *tet*(K) were not found in the SCC element of *S. haemolyticus* SH32. The plasmid-like region of IS*431* carrying *tet*(K) existed in SCC*mec* V (5C2&5c) of *S. haemolyticus* isolate NW19A from bovine milk (KM369884) (16) (Fig. 1B). The presence of an SCC*mec* element carrying class C1 *mec* complex and *ccrA4B4* in *S. haemolyticus* 48 was similar to SCC*mec* in clinical human isolate *S. haemolyticus* 83131A (CP024809) (17) but organized in the opposite position in the SCC*mec* element (Fig. 1B).

**SCC*mec*-CI of $\psi$SCC*mec*$_{384}$-$\psi$SCC$_{ars}$ in *S. hominis* subsp. *hominis* 384.** The SCC*mec*-CI of *S. hominis* subsp. *hominis* 384 (43 kb) consisted of $\psi$SCC*mec*$_{384}$ (20 kb) and $\psi$SCC$_{ars}$ (23 kb) (Fig. 2A). $\psi$SCC*mec*$_{384}$, flanked by DR1-IR1 and DR2-R2, carried a class A *mec* complex of IS*431*-*mecA*-*mecR1*-*mecI* but lacked a *ccr* gene complex (Table S3). Integration of the genes encoding trimethoprim-resistant dihydrofolate reductase (*dfrC*) and thymidylate synthase (*thyA*) was mediated by two copies of IS*257* in this region. The *dfrC* gene showed 100% nucleotide similarity to the same gene in *S. saprophyticus* UTI-058y (GenBank accession number CP054441.1), *S. epidermidis* E73 (CP035643.1), and *S. aureus* ER04164.3 (CP030542.1). In the $\psi$SCC$_{ars}$ region, the gene for spermidine N$^1$-acetyltransferase (*speG*) and the arsenical resistance operon were identified. The small subregion between IR3 and DR3-IR4 carrying *copA*, *mco*, and *ydhK* was located downstream of $\psi$SCC$_{ars}$ (Table S3).

A comparative analysis was conducted of SCC*mec* in strain 384 versus the SCC*mec* element recently identified in *S. hominis* C34847 (GenBank accession number KU936053) (18).

**FIG 1** Legend (Continued)
*S. pseudintermedius* 57395 (HE984157.2) and (B) *S. haemolyticus* 48 (JAKUUW000000000), *S. haemolyticus* SH32 (KF006347), *S. haemolyticus* 83131A (CP024809), and *S. haemolyticus* NW19A (KM369884). The percentages represent the nucleotide sequence identities. The direct repeats (DRs) and inverted repeats (IRs) are indicated by black lines. The insertion sequences (ISs) associated with antimicrobial resistance genes are indicated by the red rectangles. The following color scheme is used: *orfX* in orange, IS*431* and *mec* gene complex in pink, *ccr* genes in red, heavy metal resistance genes in yellow, ISs in purple, virulence genes in blue, restriction modification system in green, and antimicrobial resistant genes in turquoise.

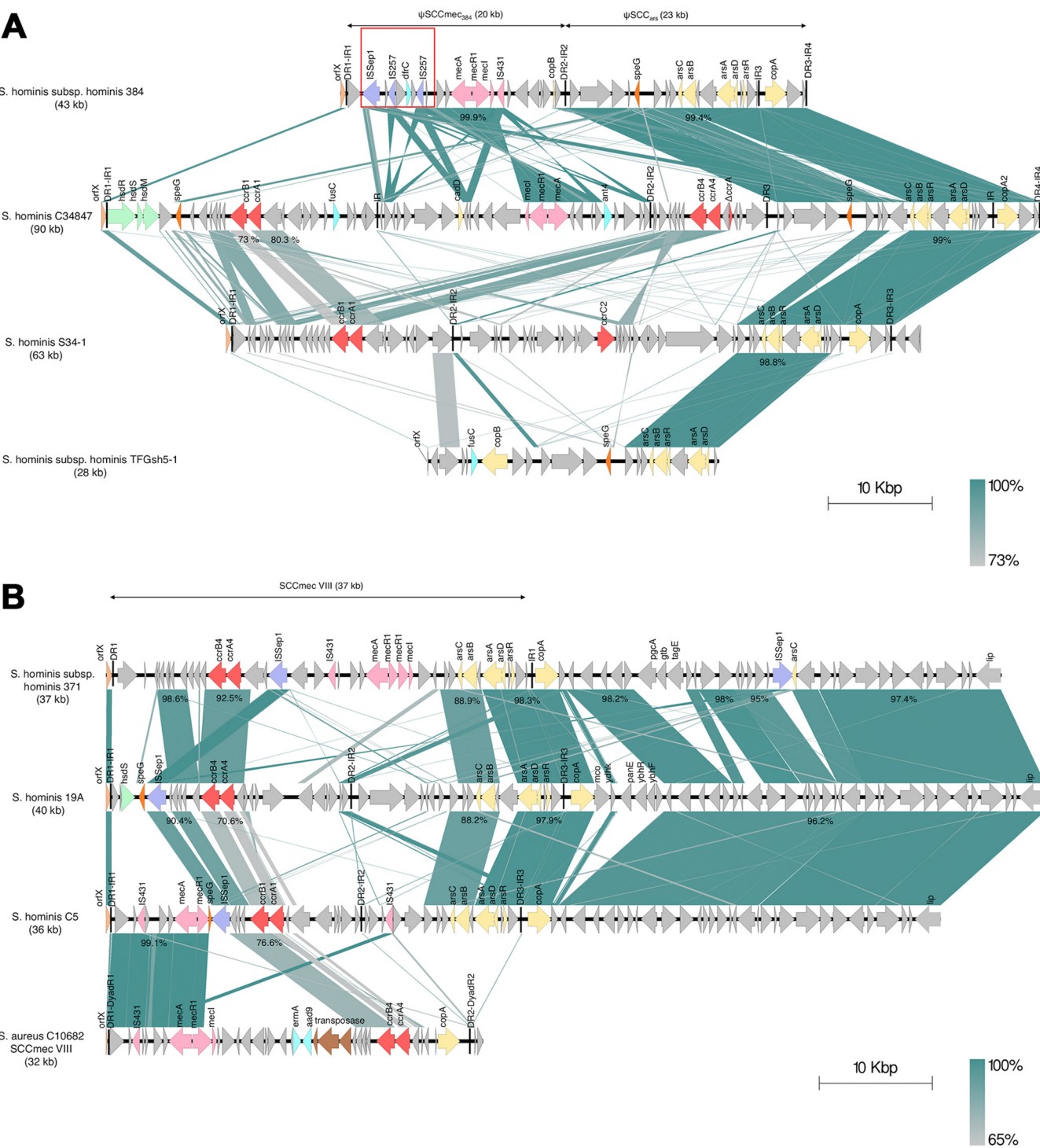

**FIG 2** Comparative analysis of the structural elements in the staphylococcal cassette chromosome *mec* (SCC*mec*) regions in (A) *S. hominis* subsp. *hominis* 384 (GenBank accession number JAKUUV000000000), *S. hominis* C34847 (KU936053), *S. hominis* S34-1 (CP040732), and *S. hominis* subsp. *hominis* TFGsh5-1 (AB930128) and (B) *S. hominis* subsp. *hominis* 371 (JAKUUU000000000), *S. hominis* 19A (NZ_CP031277), *S. hominis* C5 (CP093539), and *S. aureus* C10682 (FJ390057). The percentages represent the nucleotide sequence identities. The direct repeats (DRs) and inverted repeats (IRs) are indicated by black lines. The insertion sequences (ISs) associated with antimicrobial resistance genes are indicated by the red rectangles. The following color scheme is used: *orfX* in orange, IS*431* and *mec* gene complex in pink, *ccr* genes in red, heavy metal resistance genes in yellow, ISs in purple, virulence genes in blue, restriction modification system in green, antimicrobial resistant genes in turquoise, *speG* in dark orange, transposases in brown, and metabolic genes in gray.

The gene content and arrangement of the $\psi$SCC*mec*$_{384}$ region in strain 384 differed from those of C34847, although the $\psi$SCC*ars* and SCC*$_{384}$* regions were highly conserved (99.4%). C34847 had a class A *mec* complex like that of 384, but it was in an atypical order (Fig. 2A). The $\psi$SCC*ars* region also shared high sequence homology (>98%) and gene arrangement with $\psi$SCC-CI in the human commensal *S. hominis* S34-1 (CP040732) (19) and an SCC remnant in the human commensal isolate *S. hominis* subsp. *hominis* TFGsh5-1 (AB930128) (20) (Fig. 2A).

**SCC*mec* VIII (4A) in *S. hominis* subsp. *hominis* 371.** SCC*mec* (37 kb) harboring a class A *mec* complex and *ccrA4B4* was found in strain 371. This element was assigned to SCC*mec* VIII (4A) according to the standard guidelines (21). The genes *ccrA4* (1,629 bp) and *ccrB4* (1,362 bp) shared 100% and 99.63% nucleotide similarity to *ccrA4B4* in *S. aureus* AR466 (GenBank accession number CP029080.1), respectively. Only DR1 and an imperfect IR1 were found, and this SCC*mec* element lacked a DR at the chromosomal junction (Fig. 2B and Table S4). The *ars* operons and *copA* genes were identified near the chromosomal end of this SCC*mec* element (Fig. 2B).

According to a comparative sequence analysis, the structure of SCC*mec* VIII was highly similar to that of SCC*mec*-CI in *S. hominis* 19A isolates from buffalo milk, which had *ccrA4B4* but lacked the *mec* gene complex (22) (Fig. 2B). The chromosomal end and flanking region of SCC*mec* VIII in *S. hominis* subsp. *hominis* 371 showed high sequence homology (>88%) and gene arrangement with *S. hominis* 19A and *S. hominis* C5 (GenBank accession number CP093539) (23). *S. hominis* C5 was a human commensal isolate that possessed the new SCC*mec* type of class A *mec* complex and *ccrA1B1* (Fig. 2B). The *mec* and *ccr* gene complex in *S. hominis* subsp. *hominis* 371 was found in the opposite location from the prototype of SCC*mec* VIII in methicillin-resistant *S. aureus* (MRSA) strain C10682 (FJ390057) (24) due to gene shuffle (Fig. 2B).

## DISCUSSION

The structures of four different SCC*mec* elements in *S. haemolyticus* 1864 and 48 and *S. hominis* subsp. *hominis* 384 and 371, isolated from sick dogs, were fully characterized in this study. Analysis of the SCC*mec*-CI complex highlighted the highly mosaic nature, diverse evolutionary mechanisms, and independence of gene acquisition events in clinical *S. haemolyticus* and *S. hominis* subsp. *hominis*. The mosaic structure of their SCC*mec* elements, which were composed of genes with high similarity from various backgrounds, could imply the occurrence of gene transfer mechanisms operating within and between species, as well as in different hosts (8, 25). Mobile genetic elements on plasmids, such as antimicrobial resistance genes, allow access to a larger reservoir of niche-adaptive functions (25). One health concept considers *Staphylococcus* spp. from pets, particularly dogs, as a potential source of resistant clones and the transfer of genetic material to the human microbiome (8, 26). Infection of humans and animals by bacterial species that share a host can result from the transmission of resistant strains (26). Using PCR for SCC*mec* typing, it is difficult to detect the interchange of such elements between staphylococci of various origins. These findings support the use of full nucleotide sequencing rather than PCR designed for *S. aureus* in further research of SCC*mec* and SCC*mec* typing (3, 27).

In previous research, isolates of *S. haemolyticus* with *mecA* but lacking a *ccr* gene complex were described; it was proposed that this SCC*mec* element might be the most common type in *S. haemolyticus* (6, 14). In WCH1, the *mecA*-carrying IS*431*-formed composite transposon has been designated Tn*6191*. Five copies of IS*431* were found in WCH1, which served as a joining point and may have caused *ccr* gene deletion through homologous recombination (14). In contrast, two copies of IS*431* were identified in $\psi$SCC*mec*$_{1864}$ and $\psi$SCC*mec* of *S. haemolyticus* Sh29/312/L2; therefore, the mechanism of *ccr* gene deletion in 1864 might differ from that in WCH1. The previous study found that methicillin-resistant *S. pseudintermedius* (MRSP) carrying $\psi$SCC*mec*$_{57395}$ was predominant in Thailand (15). The presence of *mec* but lack of *ccr* in *S. haemolyticus* and *S. pseudintermedius* suggested that the *ccr* gene complex existed autonomously in these

species (15). In addition, they may have shared an ancestral SCC*mec* structure but evolved through different mechanisms.

The entire structure of SCC-CI (C1+*ccrAB4*) was first characterized in *S. haemolyticus* 48. The unique combination of the C1 *mec* complex and *ccrAB4* was previously reported in *S. haemolyticus* isolates from a patient in Bangladesh following multiplex PCR analysis (7). However, this SCC*mec* does not have a standard nomenclature. Nevertheless, five copies of IS*431* were identified, similar to those in WCH1, although *ccr* was not deleted in 48, implying that it had a different mechanism of evolution. The structure of ψSCC*mec*₄₈ was strikingly comparable to that of SCC*mec* in *S. haemolyticus* SH32 isolates from Chinese patients (5); the differences were the types of *ccr* gene complex and the insertion of IS*256* and IS*431* in 48. IS*256* is a part of the composite transposon Tn*4001*, which mediates gentamicin resistance and is more common in human clinical *S. epidermidis* strains than in commensal isolates (28). In this respect, clinical *S. haemolyticus*, which has received more exposure to antimicrobial drugs than commensal *S. haemolyticus* in animals, may be similar to human clinical *S. epidermidis*. The plasmid-like region harboring *tet*(K) on SCC*mec* was previously observed in SCC*mec* type V (5C2&5c) in *S. haemolyticus* NW19A (16) and the CI of SCC*mec* type V (Vd)+SCC*cad/ars/cop* (29). In SCC*mec*, the genome plasticity and potential to acquire antimicrobial resistance genes mediated by IS integration are highlighted by the structural and genetic diversity of the element.

An SCC*mec*-carrying class A *mec* complex lacking *ccr* was observed in *S. hominis* subsp. *hominis* 384. Elements with a class A *mec* complex but lacking *ccr* were found at 6% and 29% in clinical *S. hominis* isolates in Tunisia and Mexico, respectively (12, 27), although the complete sequences were not examined in these cases. IS*257*-mediated plasmid and transposon Tn*4003* recombination has been previously found in *S. aureus* and many CoNS, and this has been linked to trimethoprim resistance genes (*dfr*) (30). Due to the high degree of sequence similarity among *S. saprophyticus*, *S. epidermidis*, and *S. aureus*, it was assumed that *dfrC* in 384 was acquired from a non-*S. hominis* species.

SCC*mec* type VIII (4A) was identified in *S. hominis* subsp. *hominis* 371. In *S. hominis*, the prevalence of SCC*mec* type VIII (4A) has been previously reported as 10% (31), 15% (12), and 19% (32). However, there was less information on the gene arrangement and content in SCC*mec* VIII in *S. hominis*. The SCC*mec* element in strain 371 lacking DR at the chromosomal part could have been caused by homologous recombination (5). Other regions that were well conserved in strains 371, 19A, and C5 indicated the loss of the *mec* gene complex in 19A (5). It was suggested that SCC*mec* type VIII in MRSA C10682 derived from the homologous recombination between *S. aureus* and CoNS (24). The finding of loss and shared SCC structure between staphylococci supported the ability of SCC elements to be transfered and possibly evolve to a new type.

Multiple ISs may be involved in the variation of SCC*mec*-CIs in *S. haemolyticus* and *S. hominis* subsp. *hominis*, which is linked to genomic rearrangement, phenotypic diversification, and the acquisition of antimicrobial resistance (33, 34). Antimicrobial resistance genes, including aminoglycoside, tetracycline, and trimethoprim-sulfamethoxazole resistance genes coexisting on the SCC*mec* element, are important because they may limit the therapeutic options available in veterinary medicine. Related medications are commonly used to treat small animals in Thailand; therefore, coselection with SCC*mec* might have occurred in 48 and 384 (22). The presence of heavy metal resistance genes on SCC*mec*, including zinc, copper, arsenic, and cadmium resistance genes, suggests that they may have originated in CoNS (16). These pathogens could serve as a reservoir for a wide range of SCC*mec* elements and antimicrobial and heavy metal resistance genes, which could spread through clinical CoNS isolates from animals. This study described the distinct gene organization of SCC*mec* elements in clinical *S. haemolyticus* and *S. hominis* subsp. *hominis* isolates from dogs, which should help with further typing and provide better knowledge of evolution mechanisms. Due to their clinical importance, *S. haemolyticus* and *S. hominis* subsp. *hominis* from dogs may become a concern, and they may serve as a reservoir of genetic material and resistant strains for more pathogenic bacteria in close contact humans. Nonetheless, this study only looked at a small number of isolates. Continued investigation and

identification of novel SCC*mec* elements in clinical *S. haemolyticus* and *S. hominis* subsp. *hominis* using whole-genome sequencing will ensure that they are properly detected and monitored.

## MATERIALS AND METHODS

**Bacterial isolates.** This investigation contained two clinical *S. haemolyticus* and two clinical *S. hominis* subsp. *hominis* strains. *Staphylococcus haemolyticus* 1864 was isolated from the urine of a sick dog, whereas *S. haemolyticus* 48 was obtained from a skin wound of a dog in 2018. *Staphylococcus hominis* subsp. *hominis* 384 was recovered from a skin wound of a dog, and *S. hominis* subsp. *hominis* 371 was isolated from the abdominal cavity of a dog in 2017. All isolates were collected by veterinarians at a veterinary hospital and analyzed at the Veterinary Diagnostic Laboratory, Faculty of Veterinary Science, Chulalongkorn University, Thailand, using standard microbiological methods. Genetic analysis and classification as NT-SCC*mec* were performed as described in a previous study (11).

**Whole-genome sequencing and analysis.** Genomic DNA was extracted from the four strains using DNeasy blood and tissue kits (Qiagen, Hilden, Germany), following the manufacturer's instructions. The genomes were first sequenced using an Illumina NovaSeq 6000 instrument, which yielded 150-bp paired-end reads, and sequencing libraries were produced using a NEB Next Ultra II DNA library prep kit (Illumina, San Diego, USA). In addition, the genomes of the four strains were sent for long-read sequencing to obtain the entire sequences. Genomic DNA was extracted using a ZymoBIOMICS DNA miniprep kit (Zymo Research, USA) according to the manufacturer's instructions for Gram-positive bacteria. The MinION platform (Oxford Nanopore Technologies [ONT], UK) was used to sequence the genomes. The libraries were prepared using a rapid barcoding kit 004 (SQK-RBK004) and loaded onto a MinION R9.4.1 flow cell (ONT), after which they were sequenced using a MinION Mk1C device (ONT).

The Illumina raw reads were processed using fastp 0.20.1 (35), and *de novo* assembly was performed using SPAdes 3.15.2 (36). Where the contig size was >200 bp, contigs were used for gene annotation in Prokka (37). The long-read sequences were trimmed using Porechop (38). The NanoPlot tool (http://nanoplot.bioinf.be) was used to assess the quality. Read assembly and read mapping were performed using Unicycler (39), and gene annotation was conducted using DFAST (40) and Prokka (37). The structural and genetic content of SCC*mec* was manually examined. Figures showing the structural elements and comparative analyses of SCC*mec*-CIs were generated using Easyfig 2.2.2.

**Data availability.** This whole-genome shotgun project has been deposited at DDBJ/ENA/GenBank under the accession numbers JAKUUX000000000 (1864), JAKUUW000000000 (48), JAKUUV000000000 (384), and JAKUUU000000000 (371). The versions described in this paper are the first versions. The raw data are available under BioProject accession number PRJNA808897.

## SUPPLEMENTAL MATERIAL

Supplemental material is available online only.
**SUPPLEMENTAL FILE 1**, PDF file, 0.3 MB.

## ACKNOWLEDGMENT

This project was supported by a TRF Research Team Promotion Grant (RTA, 2022, Thailand), handled by Roongroj Thanawongnuwech.

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
