## [Reviewer comments · Microbiology Spectrum]

Microbiology Spectrum

Novel organization of the staphylococcal cassette chromosome mec composite island in clinical *Staphylococcus haemolyticus* and *Staphylococcus hominis* subspecies *hominis* isolates from dogs

Nathita Phumthanakorn, Thidathip Wongsurawat, Piroon Jenjaroenpun, Alongkorn Kurilung, and Nuvee Prapasarakul

Corresponding Author(s): Nuvee Prapasarakul, Chulalongkorn University

Review Timeline:

Submission Date:	March 28, 2022
Editorial Decision:	May 6, 2022
Revision Received:	June 8, 2022
Accepted:	June 16, 2022

Editor: Cezar Khursigara

Reviewer(s): Disclosure of reviewer identity is with reference to reviewer comments included in decision letter(s). The following individuals involved in review of your submission have agreed to reveal their identity: Ciro César Rossi (Reviewer #1)

Transaction Report:

DOI: <https://doi.org/10.1128/spectrum.00997-22>

May 6, 2022

Prof. Nuvee Prapasarakul
Chulalongkorn University
Department of Veterinary Microbiology
Henri-dunent road
Bangkok
Thailand

Re: Spectrum00997-22 (Novel organization of the staphylococcal cassette chromosome mec composite island in clinical *Staphylococcus haemolyticus* and *Staphylococcus hominis* subspecies *hominis* isolates from dogs)

Dear Prof. Nuvee Prapasarakul:

Two expert reviewers have examined your manuscript, and as you can see they are divided on their opinion of the work. Based on their comments, I will consider a revised submission, but ask that you pay very careful attention to the comments and address them appropriately. Specifically, Reviewer 2 rightfully points out that concerns about the description of SCC structures of non-typable and minor strains. Please respond and revise the manuscript based on all of the comments from both reviewers prior to resubmitting.

Link Not Available

Sincerely,

Cezar Khursigara

Journals Department
Reviewer comments:

Reviewer #1 (Comments for the Author):

In this work, the authors described four novel SCCmec elements, sequenced from four different strains of *S. haemolyticus* and *S. hominis* causing different types of infection in domestic dogs.

I believe that the description given here is valuable and should be published in this journal. However, I suggest some minor adjustments so that the data obtained meet its actual importance (at least in my opinion). The authors are studying two staphylococcal species that present a crescent importance as opportunistic pathogens of humans (especially *S. haemolyticus*), thus not only this study describes new variation of SCCmec, but also the possibility of this element to circulate between hosts. Which is likely to have occurred, since the sequences of the SCCmec point to a mosaicism of mobile genetic elements, including sequences found in clinical strains from humans. Also, the fact that SCCmec characterization today is mostly done by multiplex PCR could underestimate the exchange of such elements between staphylococci of different origins, as described in the following articles: 10.3389/fmicb.2017.01545 and 10.1016/j.vetmic.2019.04.009, and quickly cited in the discussion.

Minor suggestions:

It seems that the SCCmec described here are a mosaic of other mobile genetic elements described for other staphylococci. I think this mosaicism could be further discussed, which is similar to what happens to staphylococcal plasmids and probably reflects the environmental and host requirements of the microorganism isolated. This is superficially cited in line 188.

In addition, what are the origins of the *Staphylococcus* spp. to which the SCCmec described here are so similar? If they're not canine, why not discussing a bit about one health and the exchange of genetic material between staphylococcal species of different origins and hosts? I suggest taking a look at the reference doi:10.1590/1678-4685-GMB-2019-0065

Line 49: "causes MRS" sounds a bit strange...

Line 120: please specify what "in the region of SCC..." really means. Is it within the SCCmec or adjacent to it? And what does it mean?

Line 129: I miss a bit more of description of the differences between the SCCmec. In figure 1A, the SCCmec1984 and WCH1 don't look "almost identical" to me. I think some numbers (% identity) and the most important different genes (if any) could be more explored. Which are the IS that distinguish one SCCmec from the other? I understand that some of this information can be seen in Fig. 1A, but for the reader, it would be better if it was clarified, and highlighted in the figure. Some details given in the discussion seem like a description of results (like in lines 198-201). Why isn't it here?

Line 134: "The SCCmec element in *S. haemolyticus* 48..."

Line 151: Again, how "highly" similar? What is the identity?

Line 157: "The SCCmec element in *S. hominis* 384..."

Line 214: human patient?

Line 219: "clinical" is referring to a human isolate?

Reviewer #2 (Comments for the Author):

The authors have described some new findings for SCCmec and SCC structures among *S. haemolyticus* and *S. hominis* isolated from dogs.

There are some major concerns.

- The importance of these approaches should be mentioned because many of the previous articles were published nearly 10 years ago. Classification could contribute to understand the clonal expansion for problematic strains, but this study reports only 4 staphylococcal strains of 2 different species of non-typable strains. Although this type of description in this manuscript is helpful, but I think that the description of SCC structures of non-typable and minor strains might not be enough to be reported as an article. Please consider the importance of this study as a published article.

- The authors described the SCCmec-CI of these isolates are complex, but the meaning of these complexities might be based much part on assumptions. More similar isolates or clinical meanings might be needed. Some parts for these descriptions seem to be hard for the readers to understand.

- The integration of many metabolic genes in SCC region is quite interesting. For example, lipase is an important pathogenic factor for staphylococcus strains. Please check the whole genome sequence of this *S. hominis* 371 strain, and the presence of lipase in SCC region should be checked again. And more discussion for the comparison with other staphylococcal strains might be needed.

-

Minor comments

- The definition of nontypable (NT) is needed for the readers.

- The description of 19A strains is not sufficient. More information on the isolate is needed.

- L251-L258. Please describe more on the importance of this study.

Staff Comments:

Preparing Revision Guidelines

Please return the manuscript within 60 days; if you cannot complete the modification within this time period, please contact me. If you do not wish to modify the manuscript and prefer to submit it to another journal, please notify me of your decision immediately so that the manuscript may be formally withdrawn from consideration by Microbiology Spectrum.

Response to reviewers

Dear editor of Microbiology Spectrum

Herewith please find the responses to the comments of our manuscript entitled “Novel organization of the staphylococcal cassette chromosome mec composite island in clinical *Staphylococcus haemolyticus* and *Staphylococcus hominis* subspecies *hominis* isolates from dogs” follow reviewer recommendation”. We have carefully responded to all the reviewers' recommendations and modified the manuscript accordingly.

Reviewer comments:

Reviewer #1 (Comments for the Author):

In this work, the authors described four novel SCCmec elements, sequenced from four different strains of *S. haemolyticus* and *S. hominis* causing different types of infection in domestic dogs.

I believe that the description given here is valuable and should be published in this journal.

However, I suggest some minor adjustments so that the data obtained meet its actual importance (at least in my opinion).

The authors are studying two staphylococcal species that present a crescent importance as opportunistic pathogens of humans (especially *S. haemolyticus*), thus not only this study describes new variation of SCCmecs, but also the possibility of this element to circulate between hosts. Which is likely to have occurred, since the sequences of the SCCmec point to a mosaicism of mobile genetic elements, including sequences found in clinical strains from humans. Also, the fact that SCCmec characterization today is mostly done by multiplex PCR could underestimate the exchange of such elements between staphylococci of different origins, as described in the following articles: 10.3389/fmicb.2017.01545 and 10.1016/j.vetmic.2019.04.009, and quickly cited in the discussion.

Response Thank you so much for your great advice. As mentioned below, we incorporate this helpful information into the introduction and discussion sections.

Line 81-83: The transfer of genetic materials within SCCmec between human and animal strains, including as antimicrobial resistance genes and CRISPR/Cas system genes, was also detected (7,8).

Line 234-236: Their mosaic structure of SCCmec, which was composed of genes with high similarity from various backgrounds, could imply the occurrence of gene transfer mechanisms operating within and between species as well as in different hosts (8, 31).

Line 241-243: SCCmec typing by PCR is difficult to detect the interchange of such elements between staphylococci of various origins.

Reference

7. Rossi CC, Souza-Silva T, Araújo-Alves AV, Giambiagi-deMarval M. 2017. CRISPR-Cas systems features and the gene-reservoir role of coagulase-negative staphylococci. *Front Microbiol* 8:1545.

8. Rossi CC, Andrade-Oliveira AL, Giambiagi-deMarval M. 2019. CRISPR tracking reveals global spreading of antimicrobial resistance genes by *Staphylococcus* of canine origin. *Vet Microbiol* 232:65-69.

Minor suggestions:

- It seems that the SCCmecs described here are a mosaic of other mobile genetic elements described for other staphylococci. I think this mosaicism could be further discussed, which is similar to what happens to staphylococcal plasmids and probably reflects the environmental and host requirements of the microorganism isolated. This is superficially cited in line 188.

Response Thank you for your suggestion. This subject is being discussed, therefore I will take your advice. Line 234-238: Their mosaic structure of SCC*mec*, which was composed of genes with high similarity from various backgrounds, could imply the presence of gene transfer mechanisms operating within and between species as well as in different hosts (8, 31). Mobile genetic elements on plasmids such as antimicrobial resistance genes, allow access to a larger reservoir of niche-adaptive functions (31).

- In addition, what are the origins of the *Staphylococcus* spp. to which the SCCmecs described here are so similar? If they're not canine, why not discussing a bit about one health and the exchange of genetic material between staphylococcal species of different origins and hosts? I suggest taking a look at the reference doi:10.1590/1678-4685-GMB-2019-0065

Response Thank you for your suggestion and the citation. The data is incorporated into the discussion section. Line 238-241: One health concept considers *Staphylococcus* spp. from pets, particularly dogs, as a potential source of resistant clones and the transfer of genetic material to the human microbiome (8, 32). Infection of humans and animals by bacterial species that share one of the hosts can result from the transmission of resistant strains (32).

Reference

32. Rossi CC, Pereira MF, Giambiagi-deMarval M. 2020. Underrated *Staphylococcus* species and their role in antimicrobial resistance spreading. *Genet Mol Biol*. 43:1(suppl 2):e20190065.

- Line 49: "causes MRS" sounds a bit strange...

Response The sentence was rephrased.

Line 71-73: The acquisition of the *mec* gene on the staphylococcal cassette chromosome *mec* (SCC*mec*) confers MRS, which has been linked to multiple antimicrobial resistance events (1).

- Line 120: please specify what "in the region of SCC..." really means. Is it within the SCCmec or adjacent to it? And what does it mean?

Response Sorry for the mix-up. For comparison, we've updated and added additional strains. The sentence has been changed to reflect the new finding.

Line 140-149: In *S. haemolyticus* 1864, the CI of ψ SCC_{mec1864} (13 kb) and ψ SCC₁₈₆₄ (11 kb) were discovered. The ψ SCC_{mec1864} element was located immediately downstream of *orfX* and was flanked by direct repeats (DRs) and their inverted repeat (IR) sequences (DR1-IR1 and DR2-IR2) (Fig. 1A). At this position, the class C1 *mec* complex of insertion sequence (IS) *431*-*mecA*- Δ *mecR1*-IS*431*, with two copies of IS*431* orientated in the same direction, was discovered. The *ccr* gene was not found. The genes *cadC*, *arsC*, *arsB*, and *copA*, which encode cadmium, arsenic, and copper resistance, respectively, were found in ψ SCC_{mec1864} (Fig. 1A and Table S1). The second region of ψ SCC₁₈₆₄ was found immediately downstream from ψ SCC_{mec1864} and flanked by DR2-IR2 and IR3. The DR3 was not found at the chromosomal end of this element. The *ISSep1*, *cadC*, *arsC*, and *arsB* genes were found on ψ SCC₁₈₆₄.

- Line 129: I miss a bit more of description of the differences between the SCCmecs. In figure 1A, the SCCmec1984 and WCH1 don't look "almost identical" to me. I think some numbers (% identity) and the most important different genes (if any) could be more explored.

Response Thank you for your feedback, and please accept our apologies for the erroneous information. In all four strains, the SCC_{mec} element was updated and reanalyzed. SCC_{mec} in *S. haemolyticus* Sh29/312/L2 was determined as the closest match to SCC_{mec} in 1864, and the location and sequences of DRs and IRs were corrected. The results, discussion, and figures have been revised as a consequence. The percentage of nucleotide homology are shown in Figs. 1 and 2.

- Which are the IS that distinguish one SCCmec from the other? I understand that some of this information can be seen in Fig. 1A, but for the reader, it would be better if it was clarified, and highlighted in the figure.

Response Thank you for your comment. The ISs associated with antimicrobial resistance that affect SCC_{mec} are highlighted in red rectangles in Fig. 1B and 2A.

- Some details given in the discussion seem like a description of results (like in lines 198-201). Why isn't it here?

Response Apologies. This section is moved from the discussion to the results section.

- Line 155-158: The ψ SCC*mec*₁₈₆₄ showed high sequence similarity to previously described ψ SCC*mec* in *S. haemolyticus* WCH1 (accession no. JQ764731.1) obtained from the blood of a patient in China (20), and ψ SCC*mec*₅₇₃₉₅ (accession no. HE984157.2) in *S. pseudintermedius* clinical isolates from dog (21).

- Line 134: "The SCC*mec* element in *S. haemolyticus* 48..."

Response "The SCC*mec* element in 48 " has been replaced by "The SCC*mec*-CI element in *S. haemolyticus* 48" (line 140)

- Line 151: Again, how "highly" similar? What is the identity?

Response The identity is added.

Line 163-166: The gene homology and organization of ψ SCC*mec*₄₈ were highly identical (99.9%) to that of the SCC*mec* element in *S. haemolyticus* SH32 (accession no. KF006347.1), a clinical isolate from human (5) (Fig. 1B).

- Line 157: "The SCC*mec* element in *S. hominis* 384..."

Response The sentence is changed.

194-195: The SCC*mec*-CI of *S. hominis* subsp. *hominis* 384 (43 kb) consisted of ψ SCC*mec*₃₈₄ (20 kb) and ψ SCC_{ars} (23 kb) (Fig. 2A).

- Line 214: human patient?

Response The information is added.

Line 262-263: "The structure of ψ SCC*mec*₄₈ was strikingly comparable to that of SCC*mec* in *S. haemolyticus* SH32 isolates from Chinese patient (5);"

- Line 219: "clinical" is referring to a human isolate?

Response We apologize for the lack of information. The data has been added.

Line 265-266: IS256 is a part of the composite transposon Tn4001, which mediates gentamicin resistance, and is more common in human clinical *S. epidermidis* strains than in commensal isolates (34).

Reviewer #2 (Comments for the Author):

The authors have described some new finding for SCC*mec* and SCC structures among *S. haemolyticus* and *S.*

hominis isolated from dogs.

There are some major concerns.

- The importance of these approaches should be mentioned because many of the previous article were published nearly 10 years ago. Classification could contribute to understand the clonal expansion for problematic strains, but this study reports only 4 staphylococcal strains of 2 different species of non-typable strains. Although this type of description in this manuscript is helpful, but I think that the description of SCC structures of non-typable and minor strains might not be enough to be reported as an article. Please consider the importance of this study as a published article.

Response Thank you for your input and suggestions. We rewrote the text, reanalyzed the data, and amended it based on your suggestions. In the revised manuscript, the importance and necessity of characterization are discussed.

Line 28-30: SCC*mec* element distribution in clinical isolates from dogs was analyzed using whole genome sequencing and analysis, which revealed the presence of different SCC*mec* composite islands (CIs) and gene structure.

Line 81-83: The transfer of genetic materials within SCC*mec* between human and animal strains, including as antimicrobial resistance genes and CRISPR/Cas system genes, was also detected (7, 8). Line 234-243: Their mosaic structure of SCC*mec*, which was composed of genes with high similarity from various backgrounds, could imply the occurrence of gene transfer mechanisms operating within and between species as well as in different hosts (8, 31). Mobile genetic elements on plasmids such as antimicrobial resistance genes, allow access to a larger reservoir of niche-adaptive functions (31). One health concept considers *Staphylococcus* spp. from pets, particularly dogs, as a potential source of resistant clones and the transfer of genetic material to the human microbiome (8, 32). Infection of humans and animals by bacterial species that share one of the host can result from the transmission of resistant strains (32). SCC*mec* typing by PCR is difficult to detect the interchange of such elements between staphylococci of various origins.

- The authors described the SCC*mec*-CI of these isolates are complex, but the meaning of these complexities might be based much part on assumptions. More similar isolates or clinical meanings might be needed. Some parts for these descriptions seem to be hard for the readers to understand.

Response Thank you for your comments. All SCC*mec* elements have been revised and reanalyzed. We analyzed various similar SCC elements found in the Genbank database before selecting the closest homology and arrangement for the revised manuscript. The DRs and IRs that were incorrect have been fixed. Please accept

our apologies for the previous manuscript's inaccurate information. In the mark-up document, the new results, discussion and supplementary table are highlighted. Figures 1 and 2 were reworked.

- The integration of many metabolic genes in SCC region is quite interesting. For example, lipase is an important pathogenic factor for staphylococcus strains. Please check the whole genome sequence of this *S. hominis* 371 strain, and the presence of lipase in SCC region should be checked again. And more discussion for the comparison with other staphylococcal strains might be needed.

Response We apologize for the previous data. SCC*mec* in strain 371 has been reanalyzed. The *lip* gene was found outside the SCC*mec* in strain 371 and it was also detected at the similar location on the chromosome in other *S. hominis* strains. As a result, it may not be required to discuss the SCC element's relationship. Nevertheless, the *lip* gene is indicated in Fig. 2B.

Minor comments

- The definition of nontypable (NT) is needed for the readers.

Response The definition of nontypable (NT) SCC*mec* is added.

Line 89-91: The most nontypable (NT) SCC*mec* elements were found in *Staphylococcus haemolyticus* and *S. hominis* subsp. *hominis*, which were negative detection of the *mec* and/or *ccr* gene complex using multiplex PCR (11).

- The description of 19A strains is not sufficient. More information on the isolate is needed.

Response We apologize for the lack of information. The strain 19A information and reference has been added to the revised manuscript.

Line 220-222: According to comparative sequence analysis, the structure of SCC*mec* VIII was highly similar to that of SCC*mec*-CI in *S. hominis* 19A isolates from buffalo milk, which had *ccrA4B4* but lacked the *mec* gene complex (28) (Fig. 2B).

Reference

28. Pizauro LJL, de Almeida CC, Gohari IM, MacInnes JI, Zafalon LF, Kropinski AM, Varani AM. 2019. Complete genome sequences of 11 *Staphylococcus* sp. strains isolated from buffalo milk and milkers' hands. *Microbiol Resour Announc* 8(4): e01264-19.

- L251-L258. Please describe more on the importance of this study.

Response The following sentence has been included into the discussion section.

Line 301-310: This study described the distinct gene organization of SCC*mec* elements in clinical *S. haemolyticus* and *S. hominis* subsp. *hominis* isolates from dogs, which should help with further typing and provide a better knowledge of evolution mechanisms. Due to their clinical importance, *S. haemolyticus* and *S. hominis* subsp. *hominis* from dogs may become a concern and they may serve as a reservoir of genetic material and resistant strains for more pathogenic bacteria in close contact humans. Nonetheless, this study only looked at a small number of isolates. Continued investigation and identification of novel SCC*mec* elements in clinical *S. haemolyticus* and *S. hominis* subsp. *hominis* using whole genome sequencing will ensure that they are properly detected and monitored.

June 16, 2022

Prof. Nuvee Prapasarakul
Chulalongkorn University
Department of Veterinary Microbiology
Henri-dunent road
Bangkok
Thailand

Re: Spectrum00997-22R1 (Novel organization of the staphylococcal cassette chromosome mec composite island in clinical *Staphylococcus haemolyticus* and *Staphylococcus hominis* subspecies *hominis* isolates from dogs)

Dear Prof. Nuvee Prapasarakul:

Your manuscript has been accepted, and I am forwarding it to the ASM Journals Department for publication. You will be notified when your proofs are ready to be viewed.

Sincerely,

Cezar Khursigara
Editor, Microbiology Spectrum
